# Ferroelectric nematic liquids with conics

Priyanka Kumari[1,2], Bijaya Basnet[1,2], Hao Wang[1] & Oleg D. Lavrentovich ®[1,2,3] ✉

Spontaneous electric polarization of solid ferroelectrics follows aligning directions of crystallographic axes. Domains of differently oriented polarization are separated by domain walls (DWs), which are predominantly flat and run along directions dictated by the bulk translational order and the sample surfaces. Here we explore DWs in a ferroelectric nematic ($N_F$) liquid crystal, which is a fluid with polar long-range orientational order but no crystallographic axes nor facets. We demonstrate that DWs in the absence of bulk and surface aligning axes are shaped as conic sections. The conics bisect the angle between two neighboring polarization fields to avoid electric charges. The remarkable bisecting properties of conic sections, known for millennia, play a central role as intrinsic features of liquid ferroelectrics. The findings could be helpful in designing patterns of electric polarization and space charge.

Solid ferroic materials (ferroelectrics, ferromagnets, and ferroelastics) exhibit domain structures. Within each domain, the vector order parameter, such as spontaneous electric polarization **P** in ferroelectrics or magnetic moment in ferromagnets, aligns uniformly along a certain rectilinear crystallographic axis[1-4]. Domains of different orientations but of the same energy have the same probability to appear during phase transitions from a more symmetric "para" phase. The domains also form in response to a finite size of samples, to reduce depolarization fields caused by the discontinuity of the order parameter at the surfaces, as first proposed by Landau and Lifshitz[5]. Domains with a differently oriented order parameter are separated by domain walls (DWs). The ordering and thus the properties of DWs are different from those of the domains, often revealing new functionalities, such as electric conductivity in an otherwise insulating bulk, which suggests a nanotechnological potential of DWs[1-4].

DWs in solid ferroics are generally flat, as dictated by crystallographic axes and crystal facets[1-5]. The recently discovered ferroelectric nematic liquid crystal ($N_F$)[6-8] is a liquid with a macroscopic spontaneous polarization **P**, locally parallel to the director $\hat{\mathbf{n}} \equiv -\hat{\mathbf{n}}$, which specifies the average quadrupolar molecular orientation[9]. The polarization-director **P**,$\hat{\mathbf{n}}$ couple could be aligned by an approach widely used in the studies and applications of liquid crystals[10], namely, by confining the material between two glass plates with rubbed polymer coatings[7,8,11-19]. In these samples, the DW shape is defined by the anisotropic surface interactions with the "easy axis" of the substrate and by the orientational elasticity of $N_F$. The DWs in a surface-aligned $N_F$ are rectilinear[12,18], zigzag[11,15-17], lens-like[8,17], or smoothly curved[7,8,11,14,16,17,19]. Here, we explore what mechanisms shape the DWs in $N_F$ when there are no "easy axes", neither in bulk nor at the surfaces. The samples are designed with a degenerate alignment of the **P**,$\hat{\mathbf{n}}$ couple in the plane of an $N_F$ slab. We demonstrate that the DWs adopt the shape of conic sections, such as parabolas and hyperbolas. The conics bisect the angle between two neighboring polarizations in order to be electrically neutral. The $N_F$ textures avoid splay of the director but allow bend, which results in the formation of composite defects at the tips of the parabolas and hyperbolas, 180° DWs bounded by disclinations of a topological charge −1/2 each.

## Results

We explore two $N_F$ materials, abbreviated DIO[6], Figs. 1, 2, and RM734[20], Fig. 3. On cooling from the isotropic (I) phase, the phase sequence of DIO, synthesized as described previously[18], is I-174 °C-N-82 °C-SmZ$_A$-66 °C-$N_F$-34 °C-Crystal, where N is a conventional paraelectric nematic and SmZ$_A$ is an antiferroelectric smectic[17,21] (Supplementary Fig. 1). DIO films of a thickness $h = (4–10)$ μm are placed onto a surface of glycerin, an isotropic fluid. The upper surface is free (air). Both interfaces impose a degenerate in-plane alignment of the **P**,$\hat{\mathbf{n}}$ couple, as established by the measurements of DIO birefringence and thickness of the films. The phase sequence of RM734, purchased from Instec, Inc., is I-188 °C-N-133 °C-$N_F$-84 °C-Crystal. RM734 is confined between two glass plates, spin-coated with isotropic polystyrene layers in order to achieve memory-free anchoring[22]; the plates are separated by a distance $h = (1–10)$ μm. The polystyrene coatings impose a degenerate tangential anchoring of both the N and $N_F$ phases of RM734 (Fig. 3 and Supplementary Fig. 2).

[1]Advanced Materials and Liquid Crystal Institute, Kent State University, Kent, OH 44242, USA. [2]Materials Science Graduate Program, Kent State University, Kent, OH 44242, USA. [3]Department of Physics, Kent State University, Kent, OH 44242, USA. ✉e-mail: olavrent@kent.edu

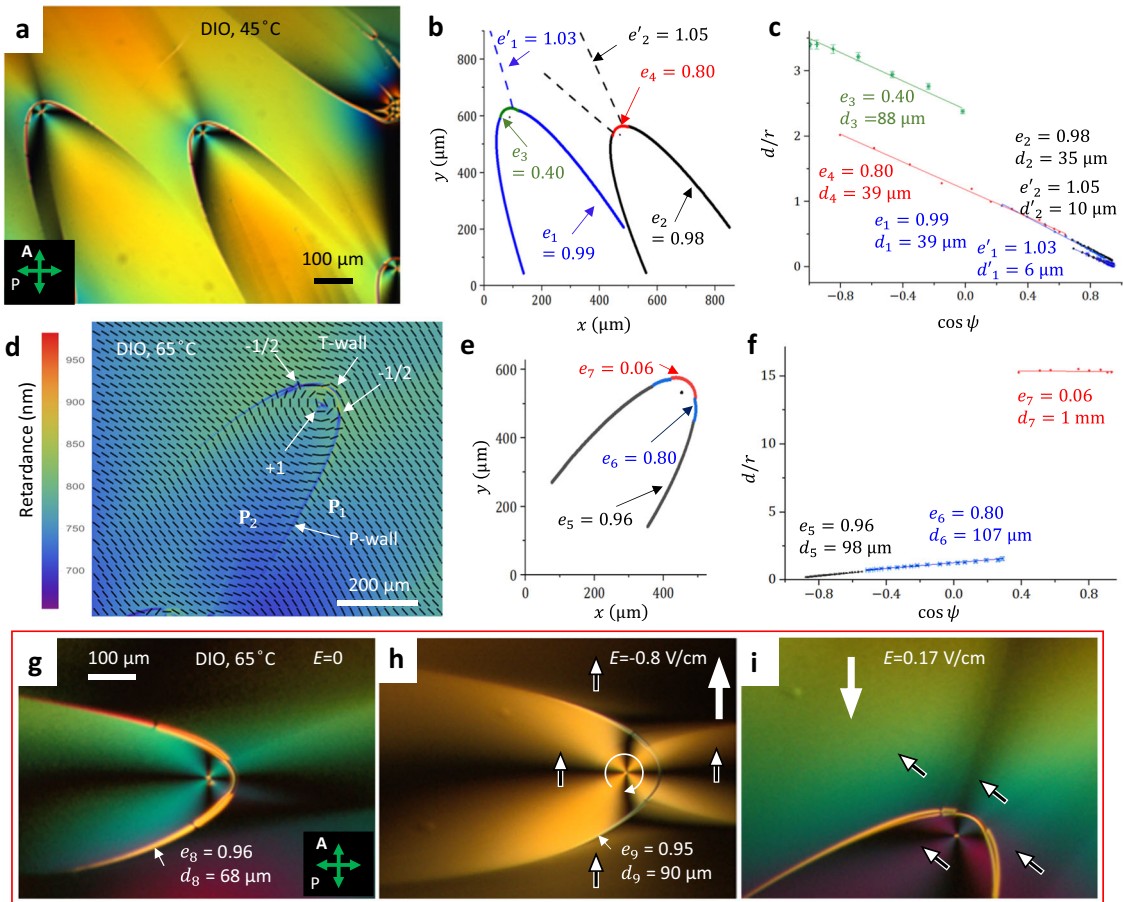

**Fig. 1 | Parabolic domain walls in $N_F$ films of DIO. a** Polarizing microscopy texture of two P-walls with foci at the cores of circular +1 vortices; the average film thickness $h \approx 4.8\,\mu m$; note two "ghost" parabolas extending to the upper left corner. **b** The corresponding eccentricities $e$. **c** The corresponding eccentricities $e$ and distances $d$ to directrices as fits to Eq. (1). **d** PolScope Microimager texture of a P-wall separating a vortex $\mathbf{P}_2$ from a nearly uniform $\mathbf{P}_1$; the tip of the P-wall is replaced by a T-wall, ending at two −1/2 disclinations; the yellow pseudo color of the T-wall is an artifact of Microimager since the actual retardance is in the lower interference order as compared to the rest of the texture; $h \approx 4.1\,\mu m$. **e, f** The corresponding $e$'s, $d$'s, and fits to Eq. (1). **g** A P-wall responds differently to the opposite polarities (**h, i**) of a dc electric field, which allows one to deduce the polarization pattern. Source data for (**b, e**) are provided as a Source Data file. The error bars in (**c**) represent the instrumental error in measuring the coordinates of the defects.

The degenerate in-plane anchoring at the $N_F$ interfaces does not require twist but permits splay and bend of $\mathbf{P},\hat{\mathbf{n}}$. However, polarizing optical microscopy textures show the prevalence of bend; the $N_F$ samples avoid splay and form domains with nearly uniform and circular director fields, as illustrated in Figs. 1 and 2 and Supplementary Figs. 3, 4 for DIO films and in Fig. 3 for flat cells of RM734. The circular director field, representing a disclination−a vortex of a topological charge +1, is directly mapped by the Pol-Scope Microimager (Hinds Instruments) in Figs. 1d and 3b. The circular vortices also manifest themselves under a conventional polarizing microscope as Maltese crosses with four extinction brushes, located in the regions where $\hat{\mathbf{n}}$ is either parallel or perpendicular to the linear polarization of incoming light, Figs. 1a, g−i, 2a, c, e, and 3a, c. Observations with a full-waveplate 550 nm optical compensator support the circular character of the director by showing interference colors of added retardance in the North-West and South−East quadrants of the Maltese cross (where $\hat{\mathbf{n}}$ is parallel to the slow axis of the compensator) and diminished retardance in the North−East and South−West quadrants (where $\hat{\mathbf{n}}$ is perpendicular to the slow axis of the compensator) (Supplementary Fig. 4).

Whenever one of the two neighboring domains in the $N_F$ textures is a circular vortex, the corresponding DW resembles a conic section. The shapes are verified with an equation of a conic, written in polar coordinates $(r,\psi)$ centered at the core of a circular vortex, as

$$\frac{d}{r} = \frac{1}{e} - \cos\psi \tag{1}$$

where $e$ is the eccentricity, $d$ is the distance from the core to the directrix. The fitted values of $e$ and $d$ are listed in Figs. 1−3; the accuracy is better than 5%. The DWs satisfy Eq. (1) with either $e \approx 1$ (parabolic, or P-walls) or $e > 1$ (hyperbolic, or H-walls) everywhere, except for the tip regions. Near the tips, the fits yield a much smaller $e$ characteristic of elliptical and circular arcs; these arcs are abbreviated as T-walls. The T-walls are 180° DWs, separating two antiparallel polarizations and bounded by two −1/2 disclinations. Besides the P-, H-, and T-walls, we also distinguish rectilinear or slightly curved B-walls (with a weak bend of the $\mathbf{P},\hat{\mathbf{n}}$ couple) that separate two closely oriented polarization fields, and C- walls, enclosing central parts of circular vortices. All these DW structures and the mechanisms of their formation are detailed below.

**P-wall** is a parabolic DW of eccentricity $e$ close to 1, Figs. 1, 3b, c, separating a uniform $\mathbf{P}_1 = \text{const}$ or a nearly uniform domain from a circular $\mathbf{P}_2$ vortex domain, as evidenced by the PolScope Microimager texture that maps the in-plane director $\hat{\mathbf{n}}(x,y)$, Fig. 1d. The polarization pattern is established by applying an in-plane direct current (dc) electric field $\mathbf{E}$, Fig. 1g−i. One polarity of $\mathbf{E}$ does not change the texture

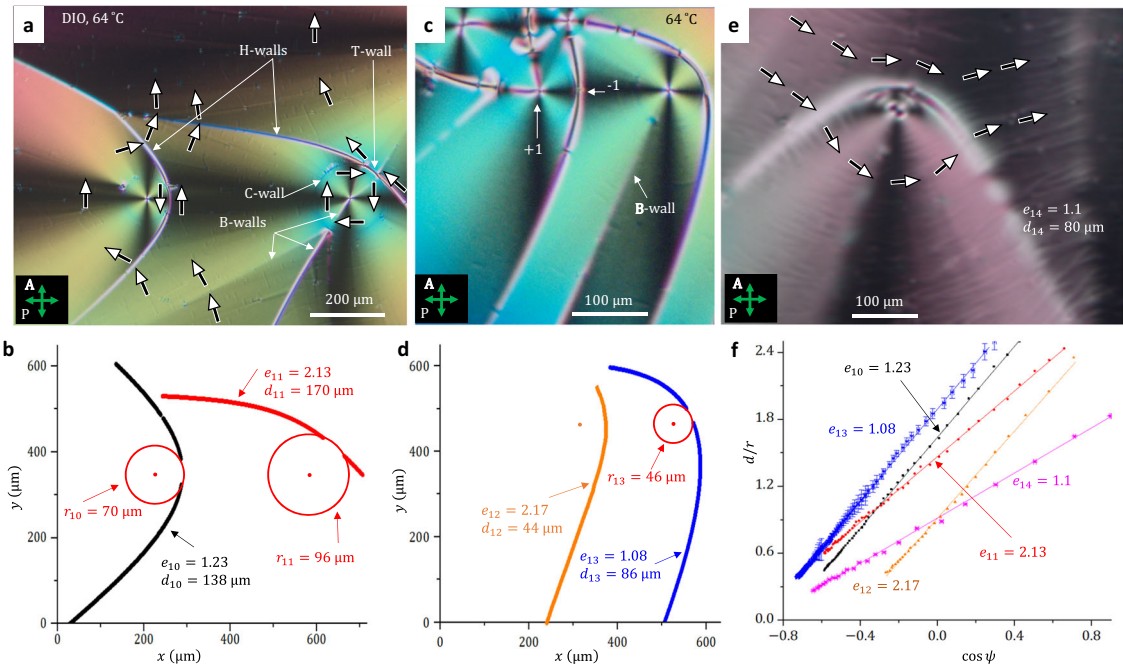

**Fig. 2 | Hyperbolic domain walls in $N_F$ films of DIO. a** Polarizing microscopy texture of two H-walls with T-walls and circular C-walls at their tips; $h \approx 6\,\mu m$; labeled by arrows are also three B-walls associated with a $+1$ circular vortex in the right-hand part of the texture. **b** The corresponding eccentricities of the H-walls in (**a**). **c** Polarizing microscopy texture of two H-wall, one of which features a $-1$ defect at the vertex with no T-wall; $h \approx 6\,\mu m$; a B-wall extends from the faint circular C-wall

to the bottom of the texture. **d** The corresponding eccentricities and distances to the directrix for the H-walls in part (**c**). **e** Polarizing microscopy texture of an H-wall in an aged sample; accumulated impurities decorate the director field; $h \approx 10\,\mu m$. **f** Fits to Eq. (1) for H-walls in parts (**a, c, e**). Source data for (**b, d**) are provided as a Source Data file. The error bars in (**f**) represent the instrumental error in measuring the coordinates of the defects.

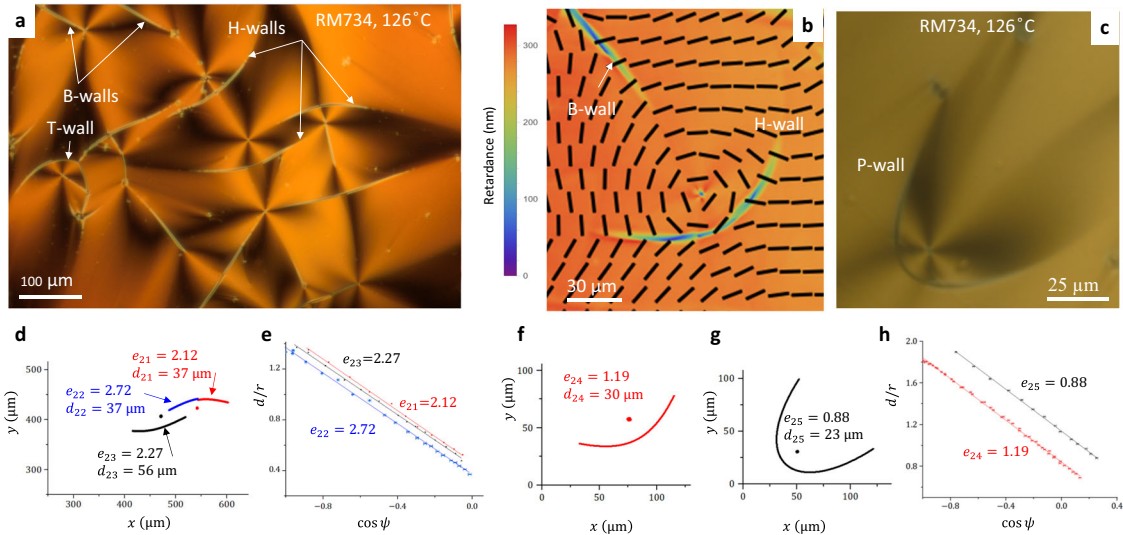

**Fig. 3 | Domain walls in flat $N_F$ cells of RM734. a** Polarizing microscopy texture with H-wall, B-walls, and T-walls; **b** PolScope Microimager texture of a B- and H-walls. **c** Polarizing microscopy texture with a P-wall. **d** Fitted branches of the H-walls in part (**a**). **e** Fitted eccentricities for the H-walls in (**a**). **f** Fitted part of the

H-wall in (**b**); **g** fitted part of the P-wall in (**c**). **h** Fitted eccentricities and distances to the directrix for the DWs in part (**b, c**). Cells thickness $h = 1.7\,\mu m$ in (**a**) and $1.3\,\mu m$ in (**b**). Source data for (**d, f, g**) are provided as a Source Data file. The error bars in (**e, h**) represent the instrumental error in measuring the coordinates of the defects.

much, Fig. 1h, while the opposite polarity reorients the parabola, Fig. 1i and Supplementary Movie 1. Therefore, $\mathbf{P}_1$ and $\mathbf{P}_2$ at the P-wall follow a head-to-tail arrangement, implying bend of $\mathbf{P}, \hat{\mathbf{n}}$, Figs. 1h and 4a, d.

The particular texture of the DIO $N_F$ film in Fig. 1a shows a variation of the interference color near the tip of the P-walls, most likely caused by the thickness gradients. Any lens-shaped film profile of a DIO film would align $\mathbf{P}, \hat{\mathbf{n}}$ orthogonally to the thickness gradient $\nabla h$ to avoid

splay; this so-called geometrical anchoring effect[23] could stabilize circular vortices. However, the geometrical anchoring is not the mechanism responsible for the occurrence of the conic sections. First, there are many DIO film areas of homogeneous interference colors and thus a uniform film thickness where the conic sections are no less ubiquitous, see Figs. 1g, h and 2c, e, Supplementary Fig. 3 and Supplementary Fig. 6c. Second, even more importantly, flat samples of

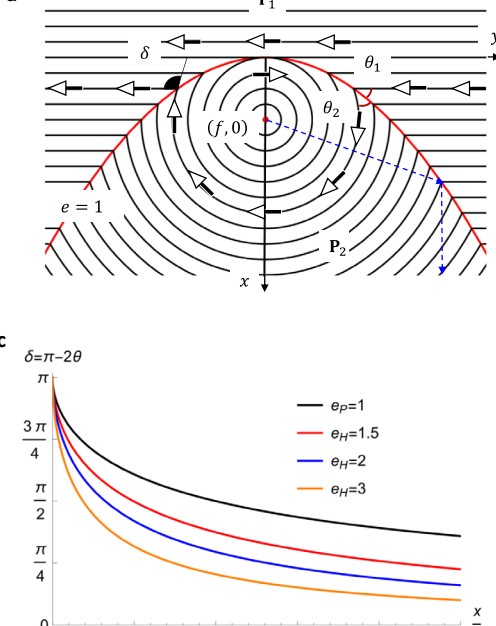

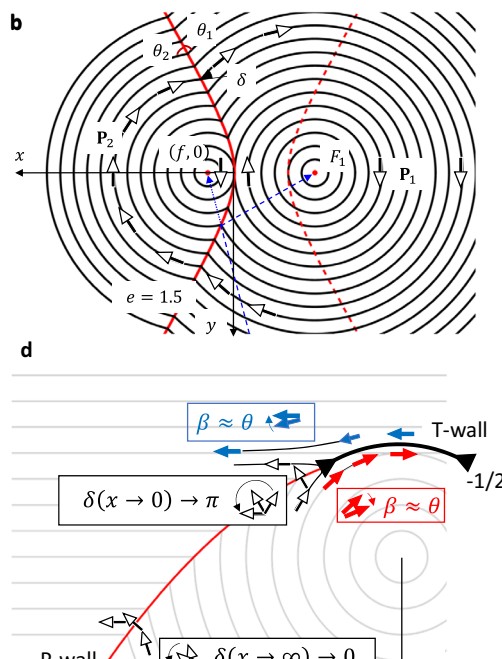

**Fig. 4 | Structure of parabolic, hyperbolic, and T-walls. a, b** Idealized P- and H-walls, respectively; the dashed blue lines illustrate reflection and bisecting properties, $\theta_1 = \theta_2$. **c** the angle $\delta = \pi\text{-}2\theta$ between $\mathbf{P}_1$ and $\mathbf{P}_2$ at the wall vs. $x/f$, calculated using Eqs. (2) and (3); $\delta \rightarrow 0$ for $x \rightarrow \infty$ and $\delta \rightarrow \pi$ for $x \rightarrow 0$. **d** At the tip of the P- and H-walls, high elastic bend energy is relieved by two −1/2 disclinations and a T-wall

joining them; each disclination replaces one large mismatch angle $\delta = \pi\text{-}2\theta$ with two small angles $\beta \approx \theta \ll \pi$, as illustrated by the boxed hodographs. The T-wall bends the outside polarization $\mathbf{P}_1$ parallel to itself. The filled black triangles represent the cores of −1/2 disclinations.

RM734 confined between glass plates with polystyrene coatings confirm unequivocally that the observed conic sections do not require any thickness gradients, Fig. 3a–c. Below we demonstrate that the shape of the P-walls is dictated by the avoidance of splay and the associated bound electric charge.

The bound electric charge in the ferroelectric bulk is of a density defined by splay, $\rho_b = -\text{div}\mathbf{P}$; the surface density of bound charge at the DWs is $\sigma_b = (\mathbf{P}_1 - \mathbf{P}_2) \cdot \hat{\mathbf{v}}_1$, where $\hat{\mathbf{v}}_1$ is the unit normal to a DW, pointing toward domain 1. Away from the cores of the DWs and the cores of circular vortices, $|\mathbf{P}_1| = |\mathbf{P}_2| = P$ since the realignments of the $\mathbf{P},\hat{\mathbf{n}}$ couple occur over length scales much larger than the molecular size. To be uncharged, a DW must bisect the angle between $\mathbf{P}_1$ and $\mathbf{P}_2$, so that $\mathbf{P}_1 \cdot \hat{\mathbf{v}}_1 = \mathbf{P}_2 \cdot \hat{\mathbf{v}}_1$.

The remarkable bisecting properties of conics, elucidated millennia ago by Apollonius of Perga[24], are often formulated in terms of light reflection[25]. Light emitted from a focus, which is the core of the circular vortex in our case, is reflected by a parabola along the lines parallel to the symmetry axis, Fig. 4a. Equivalently, a tangent to a parabola at a point $(x,y)$ makes equal angles with the radius-vector directed from the focus and with the symmetry axis. The complementary angles $\theta_1$ (between $\mathbf{P}_1$ and the P-wall) and $\theta_2$ (between $\mathbf{P}_2$ and the P-wall) are also equal, Fig. 4a, as shown in "Methods",

$$\theta_1 = \theta_2 = \arctan\sqrt{\xi} \qquad (2)$$

where $\xi = \frac{x}{f}$ and the origin of the Cartesian coordinates $(x,y)$ is at the conic's vertex. Therefore, when a P-wall separates a circular vortex of $\mathbf{P}_2$ from a uniform domain with $\mathbf{P}_1$ orthogonal to the parabola's axis, its parabolic shape guarantees that $\mathbf{P}_1 \cdot \hat{\mathbf{v}}_1 = \mathbf{P}_2 \cdot \hat{\mathbf{v}}_1$ and carries no surface charge, $\sigma_b = 0$. The bulk charge $\rho_b$ is also zero since there is no splay of $\mathbf{P}_1$ and $\mathbf{P}_2$. A small deviation of the eccentricity from $e = 1$ causes the DW between a uniform and a circular domain to carry a charge $\sigma_b \approx P(1-e)\sqrt{\frac{\xi}{\xi+1}}$, see "Methods".

The observed P-walls are not complete parabolas: they transform into a circular or elliptic conic near the tip, Fig. 1, which are the T-walls of low eccentricity as discussed later.

**H-wall** is a hyperbolic DW separating two circular vortices of the same sense of polarization circulation; $e > 1$, Figs. 2 and 3a, b, d–f. Geometrical optics is again useful to explain the reason for their existence. A hyperbolic mirror reflects a light ray aimed at one focus (the vortex core) $F_2 = (f,0)$ to the other focus $F_1$; the reflective branch of the hyperbola is between the light source and $F_2$, Fig. 4b. The tangent to a hyperbola is a bisector of the lines drawn from $F_1$ and $F_2$; the property was established by Apollonius[24]. The circular polarizations $\mathbf{P}_1$ and $\mathbf{P}_2$, which are perpendicular to the radial lines emanating from $F_1$ and $F_2$, make equal angles with the H-wall,

$$\theta_1 = \theta_2 = \arctan\left( e\sqrt{\frac{\xi^2(e-1)+2\xi}{e+1}} \right) \qquad (3)$$

see Fig. 4b and "Methods". Therefore, $\mathbf{P}_1 \cdot \hat{\mathbf{v}}_1 = \mathbf{P}_2 \cdot \hat{\mathbf{v}}_1$ and $\sigma_b = 0$ at the H-wall. If an H-wall is located midway between two vortex centers, it degenerates into a straight line, Fig. 5 and Supplementary Fig. 3.

The idealized shapes of P- and H-walls in Fig. 4a, b do not account for the fact that singular cusps of polarization at the conics are smoothed out by the $N_F$ elasticity. The width $w$ over which $\mathbf{P}$ reorients by bending is finite, increasing from a few micrometers to ~20 μm as one approaches the tip, Figs. 1g and 2a, c and Supplementary Fig. 5. The widening is caused by the increase of the misalignment angle $\delta = \pi - 2\theta$ between $\mathbf{P}_1$ and $\mathbf{P}_2$, Fig. 4c. Strong bend at the tips is relieved by disclinations of a strength −1/2 and T-walls, as discussed below.

**T-wall** is a circular or elliptical arc of low eccentricity and angular extension $0^\circ$-$180^\circ$, located at the tips of P-walls, Fig. 1a, d, g, and H-walls, Fig. 2a, c. The polarizations $\mathbf{P}_1$ and $\mathbf{P}_2$ are nearly circular, tangential to the T-wall, and antiparallel, $\mathbf{P}_1 = -\mathbf{P}_2$. Antiparallel alignment of $\mathbf{P}_1$ and $\mathbf{P}_2$ makes the T-walls similar to uncharged $180^\circ$ DWs

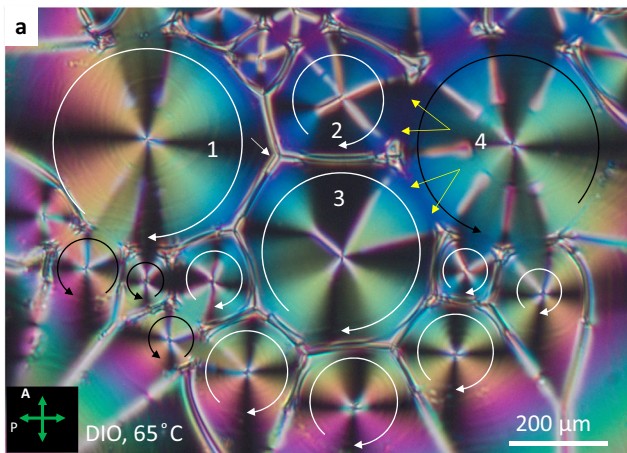
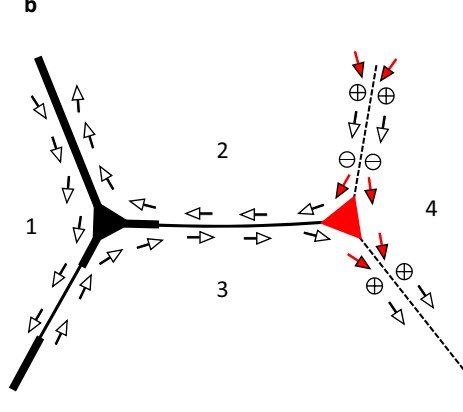

**Fig. 5 | Polygonal texture of vortices in an $N_F$ film. a** Polarizing microscopy texture. The film thickness increases from the center ($h \approx 5\,\mu m$) to the periphery ($h \approx 10\,\mu m$). Two neighboring +1 vortices of the same circulation direction are separated by H- and T-walls, while two domains of an opposite circulation show a relatively smooth transition with splay regions (shown by yellow arrows). White and black circular arrows depict polarization circulations. **b** Scheme of polarization pattern in domains 1-4 in part (**a**). Red arrows of polarization exhibit splay and produce bound charges. Thick solid lines are H-wall, thin solid lines are T-walls; dashed lines show transitions between vortices of opposite circulation at which splay produces space charges.

ubiquitous in solid ferroelectrics[1-4]. The T-walls in $N_F$ are composite defects, ending at two disclinations of strength $m_1 = m_2 = -1/2$. In a paraelectric N, half-strength disclinations are permissible as isolated defects not attached to any wall defect since $\hat{\mathbf{n}} \equiv -\hat{\mathbf{n}}$ (Supplementary Fig. 2a). In $N_F$, isolated disclinations with $m = \pm 1/2$ are prohibited since such a disclination would transform **P** into -**P** when one circumnavigates around its core[26,27]. The T-walls are exactly these topologically necessitated walls that flip **P** into -**P**. T-walls could also connect two $m_1 = m_2 = +1/2$ disclinations when an $m = +1$ core of a circular vortex splits into such a pair, Fig. 2e, Supplementary Fig. 6a, b, and Supplementary Movie 1.

The $-1/2$ disclinations and T-walls are caused by the increase of the misalignment angle $\delta = \pi - 2\theta$ between two polarization vectors $\mathbf{P}_1$ and $\mathbf{P}_2$ as one approaches the tips of P- and H-walls. Across the P- and H-walls, $\mathbf{P}_1$ and $\mathbf{P}_2$ are arranged head-to-tail. According to Eqs. (2) and (3), $\delta(x \to \infty) \to 0$ far away from the tips and the elastic energy (per unit area) $F_B \sim K_3 \delta^2 / w$ of bend from $\mathbf{P}_1$ to $\mathbf{P}_2$ is low; here $K_3$ is the bend modulus. Near the vertex, however, $\delta(x \to 0) \to \pi$, Fig. 4c, and the elastic energy of a "hairpin"-like bend in Fig. 4d is high. The $-1/2$ disclinations replace a large bent angle $\delta = \pi - 2\theta$ with two small angles $\beta \approx \theta$, Fig. 4d. The bend energy is reduced, as $2\beta^2 < \delta^2$ for $\delta_{cr} > \delta_{cr,min} = \pi(\sqrt{2} - 1) \approx 75°$. The $\delta_{cr,min}$ estimate should be revised to larger values by factors such as the disclinations core energy and the energy of polarization reversal at the T-wall. In the experiments, Fig. 1a, d, g, the T-walls often form when $\delta_{cr} = 90°-150°$, i.e., not much different from the underestimated $\delta_{cr,min}$. It suggests that the energy $F_T$ of a T-wall is of the same order of magnitude as $F_B$.

The T-wall is not completely extinct under a polarizing microscope when it is parallel to the polarizer or analyzer (Supplementary Fig. 6c), which indicates that the polarization flip involves complex deformations not directly accessible to optical inspection. The T-wall bends the $\mathbf{P}_1, \hat{\mathbf{n}}_1$ couple parallel to itself, Figs. 1d and 2e; this bend produces "ghost" diffuse conics accompanying the "proper" P- and H-walls, but extended in the opposite direction, Fig. 1a and Supplementary Fig. 3.

T-walls separate vortices of the same sense of circulation, Fig. 4b. Two neighboring vortices of the opposite sense require a polarization splay at the line of contact but no T-wall since the orientations of $\mathbf{P}_1$ and $\mathbf{P}_2$ are close to each other, Fig. 5. The associated bound charge could be estimated as $\rho_b \sim \frac{\delta P}{L} \sim (1-2) \times 10^3\,C/m^3$, where $\delta \sim \left(\frac{\pi}{4} - \frac{\pi}{10}\right)$ is the (maximum) misalignment of $\mathbf{P}_1$ and $\mathbf{P}_2$, $P = 4.4 \times 10^{-2}\,C/m^2$ is the DIO polarization density[6] and $L \geq 20\,\mu m$ is the typical length scale of splay, Fig. 5. Mobile ionic impurities of a charge density $\rho_f \sim en \sim 10^3\,C/m^3$,

where $e = 1.6 \times 10^{-19}$ C is the elementary charge, should screen $\rho_b$ if the concentration of ions in the splay regions is $n \sim 10^{22}/m^3$; the latter condition is achievable since the typical volume-averaged concentration of ions in nematics is[28] $n \sim (10^{20} - 10^{22})/m^3$. The nodes of the DW network in Fig. 5a are $-1/2$ disclinations, comprised either of three H-walls (between domains 1, 2, and 3) or one H-wall and two splay regions with bound charge (between domains 2, 3, and 4), Fig. 5b.

**B-wall** is a wall of polarization bend that separates two polarization fields of close orientation. This class embraces P- and H-walls, their ghost conics, and also DWs that form inside the vortices and between two uniform polarization domains. For example, two straight radial B-walls emanate from the core of a $\mathbf{P}_2$ vortex of the right-hand-side hyperbola in Fig. 2a; another nearly straight B-wall starts at some distance from the core of this $\mathbf{P}_2$ vortex. The B-walls are also produced by $-1/2$ disclinations in Fig. 2a. The B-walls are either rectilinear, Figs. 2a, 3a, b, or develop into conics, forming an intricate network with a branch of one DW ending at the focus of the other (Supplementary Fig. 3).

**C-wall** is a circular or slightly elliptical DW of a radius $r_{cr}$ in the range $30-120\,\mu m$, comprised of a T-wall at the tip and a very thin wall inside the P- or H-conic, centered at the core of the vortex. The polarization patterns at $r > r_{cr}$ and $r < r_{cr}$ are slightly different, for example, because of the different number of B-walls inside and outside $r_{cr}$ or because of the tendency of the $\mathbf{P}, \hat{\mathbf{n}}$ couple to partially "escape into the third dimension" by slightly tilting towards the normal $z$ within the strongly bent region $r < r_{cr}$. The latter scenario implies the appearance of twist and transformation of a cylindrical vortex towards a bend-twist structure of a Hopfion, discussed recently by Luk'yanchuk et. al. for solid ferromagnets[29]. The C-wall might thus be caused by a difference in elastic stresses in the $r < r_{cr}$ and $r > r_{cr}$ regions. Since its width is extremely narrow, the detailed structure should be explored by means other than optical microscopy.

The observed P-, H-, and B-DWs are 3D objects of a discernable length and width (on the order of $10\,\mu m$) in the plane $(x, y)$ of the sample; T- and C-walls are narrower, being a few micrometers wide (Supplementary Fig. 5). In other words, the DWs are three-dimensional objects as their width is either larger or comparable to the $N_F$ slab's thickness $h$. An interesting question is whether the DW structure changes substantially along the normal $z$ to the sample. We explore the issue by imposing shear onto the textures, namely, shifting the glass plates of the RM734 $N_F$ cell with polystyrene coatings (Supplementary Movie 2). The shear causes the

DWs to shift and extend along the shear direction but they do not split into distinct disclination "lines", which means that the structures preserve the wall character along the $z$ direction. Similarly, the +1 disclination cores of vortices do not split into two surface point defects. Some variation of the DW and +1 disclinations along the $z$ direction is expected, as in-plane deformations might trigger out-of-plane distortions; known examples are the so-called splay canceling[30], structural twist in confined achiral nematics[31,32], and twist relaxation of bend[29] mentioned above. The submicron details of the 3D structure of DWs should be explored by means such as electron microscopy since 3D optical imaging by fluorescence confocal polarizing microscopy (FCPM)[33] and coherent anti-Stokes Raman Scattering (CARS)[34,35] are not reliable because of the high birefringence of DIO[18] and RM734 (Supplementary Fig. 2b), which defocuses the probing light beam[33].

## Discussion

To summarize the results, the DWs in $N_F$ samples with degenerate azimuthal anchoring are shaped as conic sections. The textures minimize the bound electric charge in the bulk, $\rho_b = -\text{div}\mathbf{P}$, and at the DWs, of the surface density $\sigma_b = (\mathbf{P}_1 - \mathbf{P}_2) \cdot \hat{\mathbf{v}}_1$. Parabolic and hyperbolic DWs bisect the angle between two polarizations of neighboring domains, thus guaranteeing electrical neutrality. Nonzero bound charges increase the electrostatic field energy[36] $U = \frac{1}{8\pi\varepsilon_0} \iint \frac{\text{div}\mathbf{P}(\mathbf{r})\text{div}\mathbf{P}(\mathbf{r}')}{|\mathbf{r}-\mathbf{r}'|} dV' dV$, which implies a higher splay elastic constant[21,37–39], $K_1 = K_{1,0}(1 + \lambda_D^2/\xi_P^2)$, where $K_{1,0}$ is the bare splay modulus, of the same order as the one measures in N, $\lambda_D = \sqrt{\frac{\varepsilon\varepsilon_0 k_B T}{ne^2}}$ is the Debye screening length and $\xi_P = \sqrt{\frac{\varepsilon\varepsilon_0 K_{1,0}}{P^2}}$ is the polarization penetration length; $\varepsilon_0$ is the electric constant, $\varepsilon$ is the dielectric permittivity of the material, $n$ is the concentration of ions. Since $\lambda_D > \xi_P$[8], $K_1$ in $N_F$ should be much larger than $K_{1,0}$ in N and larger than the twist $K_2$ and bend $K_3$ constants in $N_F$. Experiments on planar DIO cells[18] suggest $K_1/K_3 > 4$, in line with the observed predominance of bend in the textures of conics. Expulsion of splay is not absolute, however, since some splay develops at the border of vortices with opposite sense of polarization circulation, Fig. 5.

The title of this paper is partially borrowed from the 1910 publication[40] by G. Friedel and F. Grandjean that described ellipses and hyperbolas seen under a microscope in a liquid crystal of a type unknown at that time. A later analysis[41,42] revealed that these conics are caused by a layered structure of the liquid crystal known nowadays as a smectic A (SmA). The layers are flexible but preserve equidistance when curled in space. The normal $\hat{\mathbf{n}}$ to the equidistant layers, which is also the director, can experience only splay but not twist nor bend; the focal surfaces of families of these layers reduce to confocal conics, such as an ellipse-hyperbola or two parabolas[43]; these pairs form the frame of the celebrated focal conic domains (FCDs)[44]. Gray lines in Fig. 4a, b, could be interpreted as cuts of smectic layers wrapped around a parabola and hyperbola of FCDs. The smectic structure is stabilized by the requirement $\text{curl}\hat{\mathbf{n}} = 0$, a conjugate to the condition $\text{div}\hat{\mathbf{n}} = 0$ in $N_F$. The described $N_F$ liquid with conics is shaped by a different physical mechanism, rooted in electrostatics, namely, in the avoidance of the space charge. Electrostatics hinders splay of $\hat{\mathbf{n}}$ and $\mathbf{P}$, so that the lines of $\hat{\mathbf{n}}$ and $\mathbf{P}$ are "equidistant" (divergence-free). Besides this difference in the physical underpinnings, there is also a distinction in how the conics in $N_F$ and SmA heal cusp-like singularities. In $N_F$, the cusps are attended by a bend of the polar vector $\mathbf{P}$, which necessitates the −1/2 disclinations and the T-walls at the tips of the conics, while in a SmA, a similar cusp could be healed by weak splay of the apolar director $\hat{\mathbf{n}} \equiv -\hat{\mathbf{n}}$.

T-walls bounded by half-integer disclinations have been predicted for $N_F$[26,27] as analogs of DWs seeded by cosmic strings in the early Universe models[45] and of DWs bounded by half-quantum vortices recently found in superfluid $^3$He[46,47]. In the Universe and $^3$He scenarios, the composite DWs appear after a phase transition from a symmetric phase that contains isolated strings/disclinations. In the less symmetric phase, the isolated disclinations are topologically prohibited and must be connected by a DW. In contrast, the −1/2 disclinations at the ends of T-walls described in this work serve to reduce the elastic energy of strong bends, Fig. 4d, and appear without any reference to more symmetric phases. The detailed core structures of the observed DWs require further studies with a resolution higher than that of optical microscopy. The fine structure should include "space-charge electric double layer", i.e., two parallel sheets of positive and negative space charges produced by the increase of the projection of $\mathbf{P}$ onto $\mathbf{v}_1$ midway across the DW, when the polarization bends from $\mathbf{P}_1$ to $\mathbf{P}_2$, Fig. 4d, as discussed by Pattanaporkratana[48] for $m = -1$ disclinations in ferroelectric smectic C and by Chen et al.[49] for parabolic walls induced by air bubbles in planar $N_F$ cells. Even for relatively low polarizations, if one neglects the screening effects of ions, mutual attraction of these sheets is expected to reduce the width of director reorientation dramatically, down to ~10 nm[48]; ionic screening would expand this width. Therefore, the DW cores are shaped by an intriguing balance of the elastic, space-charge, and ionic effects at the length scales of micrometers and below.

The demonstrated interplay of electrostatics and geometry that shapes the DWs in ferroelectric fluids could be potentially explored in the design-on-demand of electric polarization and space charge. For example, surface patterning could be used to predesign a gradient director field with certain deformation types, including splay, which would generate patterns of space charge with a pre-programmed response to an externally applied electric field.

## Methods

### Sample preparation and characterization

**DIO**. The dioxane ring of DIO molecules could be in *trans*- or *cis*-conformations. The *cis*-isomer is not mesomorphic and its presence in the material could significantly decrease the transition temperatures[50]. For example, the I–N transition temperature in the *cis:trans* = 10:90 composition is 150 °C, which is 24 °C lower than the temperature 174 °C of the I–N transition of a *cis:trans* = 0:100 composition. The transition temperatures of the DIO synthesized in our laboratory are very close (within 2 °C) to the ones reported by Nishikawa et al.[50] for the composition *cis:trans* = 0:100. We conclude that the DIO studied in this work is comprised entirely of the *trans*-isomers.

The DIO films with degenerate azimuthal surface anchoring are prepared by depositing a small amount of DIO onto the surface of glycerin (Fisher Scientific, CAS No. 56-81-5 with assay percent range 99–100% w/v and density 1.26 g/cm$^3$ at 20 °C) in an open Petri dish. A piece of crystallized DIO is placed onto the surface of glycerin at room temperature, heated to 120 °C, and cooled down to the desired temperature with a rate of 5 °C/min. In the N, SmZ$_A$, and $N_F$ phases, DIO spreads over the surface and forms a film of an average thickness $h$ defined by the known deposited mass $M$ and the measured area $A$ of film, $h = M/\rho A$, where $\rho = (1.32–1.36)$ g/cm$^3$ is the density of DIO in the $N_F$ phase measured in the laboratory. For example, the film in Fig.1d was formed by 2.7 mg of DIO spread over the area $A = 4.91$ cm$^2$; with $\rho = 1.33$ g/cm$^3$ at 65 °C, its thickness is $h = \frac{M}{\rho A} \approx 4.1\,\mu$m. The temperature dependence of DIO density (Supplementary Fig. 7), is determined by placing a known amount $M$ of DIO in a flat sandwich cell of a known thickness ($h = 50\,\mu$m) and measuring the area $A$ occupied by the material, $\rho = M/Ah$.

The $h$ values are in good correspondence with the DIO film thickness determined by optical means as $h_\Gamma = \Gamma/\Delta n$, where $\Gamma$ is the optical retardance of the film either measured using PolScope Microimager (Hinds Instruments), as in Fig. 1d, or estimated from interference colors of the textures according to the Michel−Levy chart; $\Delta n = n_e - n_o$ is the birefringence; $n_o$ and $n_e$ are the ordinary and extraordinary refractive indices, respectively. The temperature dependence of DIO birefringence $\Delta n$ was measured previously at the

wavelength 535 nm[18]. At the temperatures of interest in our study, $\Delta n = 0.20$ at 45 °C and $\Delta n = 0.19$ at 65 °C. The optical retardance of the film's texture in Fig. 1d is in the range (740–780) nm; using the birefringence $\Delta n = 0.19$, one finds $h_\Gamma = (3.9–4.2)$ μm, close to $h = 4.1$ μm. Since the $h_\Gamma$ and $h$ are similar and since $\Gamma$ of DIO $N_F$ films attains its maximum possible value $h\Delta n$, the data suggest that the director and polarization **P** are tangential to the $N_F$-air and $N_F$-glycerin interfaces.

**RM734**. The material was purchased from Instec, Inc. (purity better than 99%), and additionally purified by silica gel chromatography and recrystallization in ethanol. Flat cells of RM734 are assembled from glass plates spin-coated with layers of polystyrene, separated by a distance $h = (1–10)$ μm and sealed with an epoxy glue Norland Optical Adhesive (NOA) 65. Glass substrates are cleaned ultrasonically in distilled water and isopropyl alcohol, dried at 95 °C, cooled down to room temperature and blown with nitrogen. Spin coating with the 1% solution of polystyrene in chloroform is performed for 30 s at 4000 rpm. After the spin coating, the sample is baked at 45 °C for 60 min. Two polystyrene-coated glass plates are assembled into cells and filled with RM734 by capillary force at 150 °C and cooled down to 126 °C with a rate of 5 °C/min. The cell thickness $h$ was measured by a light interferometry technique using a UV/VIS spectrometer Lambda 18 (Perkin Elmer). The textures show a degenerate tangential alignment in both N and $N_F$ phases (Fig. 3a and Supplementary Fig. 2a).

Planar cells with unidirectionally rubbed polyimide PI-2555 aligning layers are used to determine the temperature dependency of RM734 birefringence $\Delta n = \Gamma/d$ (Supplementary Fig. 2b); $\Gamma$ is measured by PolScope Microimager. At 535 nm, $\Delta n = 0.23$ at 146 °C in the N phase and $\Delta n = 0.25$ at 126 °C in the $N_F$ phase.

Textures and optical retardance of flat RM734 cells with polystyrene coatings provide direct evidence of the tangential anchoring of the director in both the N and $N_F$ phases. The N Schlieren texture of RM734 shows isolated disclinations of strength +1/2 and −1/2, which are not connected to any wall defects (Supplementary Fig. 2a). Isolated ±1/2 are possible only when the surface anchoring is strictly tangential (a tilted anchoring requires a wall, which bridges positive and negative directions of tilt, see refs. [44,51]). The retardance of the texture in a cell of a thickness $h = 1.7$ μm at 146 °C is estimated by the Michel−Levy chart as $\Gamma \approx 400$ nm (Supplementary Fig. 2a), which implies $\Delta n = \Gamma/h = 0.235$, in agreement with an independently measured birefringence (Supplementary Fig. 2b). As the temperature is lowered to 126 °C, the retardance increases to ≈440 nm (Fig. 3a, which is the same sample area as in Supplementary Fig. 2a). This increase is expected, as the birefringence in the $N_F$ phase, $\Delta n = 0.25$, is higher than in the N phase (Supplementary Fig. 2b). Since $\Gamma$ in the $N_F$ phase reaches its maximum possible value $\Gamma = h\Delta n$, one concludes that the director and polarization are tangential to the bounding plates. The same conclusion about tangential alignment of RM734 follows from Fig. 3c, in which the PolScope Microimager shows $\Gamma \approx 300$ nm at the wavelength 655 nm; with the known cell thickness $h = 1.3$ μm it yields $\Delta n = 0.23$, in agreement with the independently measured birefringence (Supplementary Fig. 2b).

The optical textures are recorded using a polarizing optical microscope Nikon Optiphot-2 with a QImaging camera and Olympus BX51 with an Amscope camera.

## Calculations of surface charge at parabolic and hyperbolic domain walls

**P-walls**. We calculate the surface charge at a DW separating a uniform $P_1$ domain and a circular polarization $P_2$. The DW could be a parabola or a conic with eccentricity $e$ somewhat different from 1. A general formula of a conic section expressed in Cartesian coordinates $(x,y)$, the origin of which coincides with the conic's vertex, writes

$$x^2(e^2 - 1) + 2xf(e + 1) - y^2 = 0 \qquad (4)$$

where $f$ is the distance between the vertex and the focus F($f$,0) of the conic. For any point M($x,y$) at the conic, the distance MF to the focus and MD to the directrix $x = -f/e$ satisfies the equality $(MF)^2 = e^2(MD)^2$. The focus F($f$,0) is a center of the circular polarization. An acute angle between two curves $y = g_1(x)$ and $y = g_2(x)$ intersecting at a point $(x,y)$ is calculated using $\tan\theta = |(s_1 - s_2)/(1 + s_1 s_2)|$, where $s_i = \frac{dg_i(x)}{dx}\big|_x$, $i = 1,2$. One finds $\theta_1 = \arctan\frac{1}{1+\xi(e-1)}\sqrt{\frac{2\xi+\xi^2(e-1)}{e+1}}$ and $\theta_2 = (\arctan)\,e\sqrt{\frac{2\xi+\xi^2(e-1)}{e+1}}$, where $\xi = x/f$. For $e = 1$,

$$\theta = \theta_1 = \theta_2 = \arctan\sqrt{\xi} \qquad (5)$$

The parabola bisects the angle between $P_1$ and $P_2$, so that $P_1 \cdot \hat{v}_1 = P_2 \cdot \hat{v}_1$, or $P\sin\theta_1 = P\sin\theta_2$, and $\sigma_b = 0$. If the DW deviates from the parabolic shape toward an ellipse or a hyperbola, but the fields $P_1$ and $P_2$ do not alter, the charge is finite, $\sigma_b = P(1 - e)\sqrt{\frac{2\xi+\xi^2(e-1)}{(e\xi+1)[1+e+e\xi(e-1)]}}$, which simplifies to $\sigma_b \approx P(1 - e)\sqrt{\frac{\xi}{\xi+1}}$ for small departures of $e$ from 1.

**H-wall**. separates two domains of circular polarization. It is convenient to use the Cartesian coordinates $(x',y)$ with the origin at the half-distance $a$ from the two vertices of the hyperbola:

$$\frac{(x')^2}{a^2} - \frac{y^2}{c^2 - a^2} = 1$$

where $(c,0)$ and $(−c,0)$ are the coordinates of the focal points, and $(a,0)$ and $(−a,0)$ are the vertices of the hyperbola. The two circular polarization fields are

$$P_2 = P\left(\frac{x' + c}{\sqrt{(x' + c)^2 + y^2}}, \frac{y}{\sqrt{(x' + c)^2 + y^2}}\right) \text{ and}$$

$$P_1 = P\left(\frac{x' - c}{\sqrt{(x' - c)^2 + y^2}}, \frac{y}{\sqrt{(x' - c)^2 + y^2}}\right)$$

Calculations similar to the one above for a **P-wall** predict that two polarizations intersect the hyperbola at the same angle,

$$\theta = \theta_1 = \theta_2 = \arctan\left(\frac{c}{a}\sqrt{\frac{(x')^2 - a^2}{c^2 - a^2}}\right)$$

which implies $\sigma_b = 0$; $\rho_b = 0$ since $P_1$ and $P_2$ are circular.

In order to compare how the tilt angles vary with the horizontal coordinate $x$ in parabola and hyperbola, the last result could be rewritten as

$$\theta = \theta_1 = \theta_2 = \arctan\left(e\sqrt{\frac{\xi^2(e - 1) + 2\xi}{e + 1}}\right) \qquad (6)$$

in the Cartesian coordinates with the origin at the vertex of the hyperbola shown by a solid red line in Fig. 3b and with the focus at $(f,0)$; $\xi = x/f$. The angle $\delta = \pi - 2\theta$ between the two vectors $P_1$ and $P_2$ then writes

$$\delta_P = 2\,\text{arccot}\sqrt{\xi};$$

$$\delta_H = 2\,\text{arccot}\left(e\sqrt{\frac{\xi^2(e - 1) + 2\xi}{e + 1}}\right)$$

for the P- and H-walls, respectively. The mismatch angle $\delta_P$ increases faster than $\delta_H$ near the conic's tip, thus the elastic bend stress is

higher at the P-walls as compared to the H-walls, Fig. 4c. Thus the angular width of the T-walls that eliminate the strong bend is generally larger at the P-walls, Fig. 1a, d, g, as compared to the H-walls, Fig. 2. In some cases, the two −1/2 disclinations at the tip of H-walls coalesce into a single −1 defect, see the left-hand-side H-wall with $e$ = 2.17 in Fig. 2c.

## Data availability
The authors declare that the data supporting the findings of this study are available within the text, including "Methods" and Extended Data files. The datasets generated during and/or analyzed during this study are available from the corresponding author on request. Source data are provided with this paper.

## Code availability
A Mathematica code is available in the Supplementary Information.

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

## Acknowledgements
We thank Noel A. Clark for illuminating discussions and for sharing Refs. [48,49] and Kamal Thapa for the help in the experiments. This work is supported by NSF grant ECCS-2122399 (O.D.L.).

## Author contributions
P.K. performed the experiments. P.K., B.B., and O.D.L. analyzed the data. H.W. synthesized the DIO material. O.D.L. conceived the idea and wrote the manuscript with input from all co-authors.

## Competing interests
The authors declare no competing interests.

## Inclusion and ethics
Research has been conducted at Kent State University and all contributors have been acknowledged.
