## [Peer Review File · Nature Communications]

REVIEWER COMMENTS

Reviewer #1 (Remarks to the Author):

This work claims the observation of domain walls in NFLCs. The authors focus on a smectic-like texture (conics) in NFLC. The authors observed the conics by a polarising microscope, determined the director field by polscope and speculated the P vector field. They used image analysis use extract the shape characteristics. Based on the speculation of the P field and the shape features of the conics, they mainly discussed the relationship between the bound electric charge and the P orientation near the domain walls. Overall, the result is not impressive and lacks crucial experimental foundations.

I have some questions on the validity of manuscript and conclusions:

1.The manuscript is rudely written. No well-organized introduction, difficult to follow.

2.The analysis is based on the speculation and not complete. Authors mentions the LC film is 3D (e.g. In Fig.1a, optical retardance at the tips of the P-walls is lower than in the rest of the film. The lens-shaped film profile forces $\mathbf{P}\hat{\mathbf{n}}$ to be orthogonal to ∇h to avoid splay; this so-called geometrical anchoring), but they made their analysis of domain walls without mentioning any details of the film geometry and defects. Wall or line? Line attached on surface, floating on bulk, or wall connecting two substrates? What is the shape of the film? Is it homogenously lens-shaped? Or lens shaped only for conics part but flat for uniform alignment area? But their model assumes the defects in 2D plane. Also, they say "The conics in the NF films are likely to be walls, since in order to shrink a wall into a line, the polarization \mathbf{P} should tilt away from the xy plane of the film." Whether it is a wall or something else can be readily tested by confocal microscopy.

Here are some more minor comments:

1.Some typos and undefined terms. e.g. "... fluid that imposes a degenerate in-plane alignment of the polarization-director $\mathbf{P}\hat{\mathbf{n}}$ couple." "The T-wall bends the $\mathbf{P}\hat{\mathbf{n}}$ couple parallel to itself," What is the physical meaning of $\mathbf{P}\hat{\mathbf{n}}$? First time to see this notation and used frequently. Should read \mathbf{P} (and $\hat{\mathbf{n}}//\mathbf{P}$), $\mathbf{P}\cdot\hat{\mathbf{n}}$?

2.Loose in references. Some important references are not cited. See above.

3. Figures should be reorganized. The authors use atypical axis labels. They should reformat the axis label to right positions.

4. Units should be written in nonitalic.

Reviewer #2 (Remarks to the Author):

"Ferronematic liquids with Conics" by Kumari et al is a beautiful piece of work, with a comprehensive coverage of the textures formed by this most exciting and current of novel materials when in contact with degenerate surface conditions. There is no hesitation that this work should be published and that Nature Communications is a good choice of journal. I

Some minor points the authors should consider are listed below.

"find their material realization.." is a little grand. Conic sections are well known to be important in smectic liquid crystals. Chiral smectic C liquid crystals are fluidic and hence may be thought of as ferrofluids. I would recommend being a little less poetic and using "play a central role as intrinsic..." or something similar.

It would be instructive to include a red-tint plate (or similar lamda plate) for (at least one) of the photomicrographs. For example, it is stated on page 1 that splay is avoided and the domains circular. This is evinced by the Polscope image in Fig 1d). However, the degeneracy should be immediately evident to the reader if the lamda plate photomicrographs were also included.

The jump to describing the different wall structures is a little severe, and left me searching through the paper when I got to the B wall. Everything had been very clear yet simple to that point. Hence, I think it would be better to soften this with an earlier mention that Fig 1 and 2 (2 in particular) have examples of five different types of wall P, H, T, B and C, which will be described in detail in following sections.

Page 1:

"..forms nearly uniform and circular domains" is not clear from the figures referenced. Presumably, it refers to the photomicrographs of these figures, such as Fig 1a), g), and h), etc. Presumably, it means that the defect at the foci retain orthogonal brushes to some distance, thereby implying circular structure of the director field. It would be helpful to the reader to clarify what is meant.

Seems odd seeing "focues" alongside "vortices" in the figure caption for Fig 1. I believe foci is preferable, but this might be a journal preference to use contemporary US English.

The labels for Fig 1 b and c are confused. Both are fits to equation 1, with b) the eccentricities and c) the directrices.

Figure 2 should mention the B wall in the caption.

Page 4: " as discussed later".

Page 9: the use of Ref before 17 is inconsistent and un-necessary. Move to before the equation if preferred.

Reviewer #3 (Remarks to the Author):

Ferroelectric nematic (NF) liquid crystals is a both a new material and new field of very intense research. It has gained large interest far outside the liquid crystal research community.

The authors of the present manuscript have studied a ferroelectric nematic liquid crystal in a novel sample geometry, using degenerate planar and slippery surfaces - glycerol (bottom) and air (top) - as confinement surfaces. With these azimuthally degenerate boundary conditions twist is not required in the LC. Further, in ferroelectric nematics, as a result of the very large value of spontaneous polarization, splay is avoided in order to minimize space charge from divergence in P . The authors have observed and explained that the resulting NF structure primarily contains pure bend deformations, with domain walls (sheets) therefore, importantly, forming conics.

The authors, in great detail, theoretically explain their experimental findings. The methods and conclusions are sound and convincing, and the paper is well written.

The authors also point out the analogy with smectic A materials where twist and bend are suppressed and the resulting structure is primarily splayed in the director, here leading to well-known focal conics in the form of disclination lines.

To summarise, this is a beautiful, original, and important study and I certainly recommend the manuscript for publication in Nature Communications.

I have only the following (minor) comments/suggestions:

1) In the abstract the authors suggest that:

“The findings could be helpful in designing patterns of electric polarization and space charge”. However, this statement is as far I can see not discussed or elaborated on in the paper. I think such a statement in the abstract should indeed be discussed further/motivated (seemingly being one conclusion from the study).

2) I think it would even be more interesting if the authors could comment on possible implications of the results for the future potential use of the NF materials in e.g. display or photonics applications.

3) The estimation of the film thickness h could be clarified in little more detail.

In caption Figure 1:

a) “average film thickness is estimated by the interference colors”.

In the text:

b) "Optical retardance Γ of NF films is close to its maximum possible value $h\Delta n$, where Δn is birefringence measured at different temperatures and wavelengths previously. Thus **P** is tangential or only slightly tilted away from the xy film plane".

In Methods:

c) "an average thickness \bar{h} defined by the deposited mass and the area of film."

My question is simply: Was h calculated from the weight and density of DIO, and the film area (c), or was h calculated from the retardation and birefringence, assuming tangential orientation of P (which is what you would expect) (a), or was the tangential orientation of P determined from the birefringence, using a value of h estimated from (c)?

4) The font size in Figures 2 and 3 could possibly be made little larger.

Dear Editors,

We are thankful to you and the Reviewers for the evaluation of our work, critical remarks, and suggestions to improve the presentation. We addressed all the critical remarks and suggestions in the revised manuscript. Below we list the comments and our point-by-point replies.

REVIEWER COMMENTS

Reviewer #1 (Remarks to the Author):

R1.0.

This work claims the observation of domain walls in NFLCs. The authors focus on a smectic-like texture (conics) in NFLC. The authors observed the conics by a polarising microscope, determined the director field by polscope and speculated the P vector field. They used image analysis use extract the shape characteristics. Based on the speculation of the P field and the shape features of the conics, they mainly discussed the relationship between the bound electric charge and the P orientation near the domain walls. Overall, the result is not impressive and lacks crucial experimental foundations.

A1.0.

We thank Reviewer 1 for the evaluation of our work. Our experiments and analysis demonstrate a novel shape of defects in ferroelectric materials and explain the physical reason for these novel shapes. In the revised manuscript, we added new experiments on the second ferroelectric nematic, RM734, placed in cells formed by rigid (glass) plates with degenerate in-plane anchoring, in order to demonstrate the generality of the original findings.

R1.1. I have some questions on the validity of manuscript and conclusions:

1.The manuscript is rudely written. No well-organized introduction, difficult to follow.

A1.1. We reorganized and expanded the Introduction to make sure that it is easy to follow.

R1.2. 2.The analysis is based on the speculation and not complete. Authors mentions the LC film is 3D (e.g. In Fig.1a, optical retardance at the tips of the P-walls is lower than in the rest of the film. The lens-shaped film profile forces $\mathbf{P}\hat{\mathbf{n}}$ to be orthogonal to ∇h to avoid splay; this so-called geometrical anchoring), but they made their analysis of domain walls without mentioning any details of the film geometry and defects. Wall or line? Line attached on surface, floating on bulk, or wall connecting two substrates? What is the shape of the film? Is it homogenously lens-shaped? Or lens shaped only for conics part but flat for uniform alignment area? But their model assumes the defects in 2D plane. Also, they say “The conics in the NF films are likely to be walls, since in order to shrink a wall into a line, the polarization \mathbf{P} should tilt away from the xy plane of the film.” Whether it is a wall or something else can be readily tested by confocal microscopy.

A1.2. The films do not need geometrical anchoring or lens shape for the conics to form. In the revised manuscript, we added new experiments, Fig. 3, which demonstrate that the conics form

when there are no thickness gradients, in flat cells formed by two glass plates. We added a text on p. 4-5:

“However, the geometrical anchoring is not the mechanism responsible for the occurrence of the conic sections. First, there are many DIO film areas of homogeneous interference colors and thus a uniform film thickness where the conic sections are no less ubiquitous, see Figs. 1g,h, 2c,e, Supplementary Fig. 3 and Fig. 6c. Second, even more importantly, flat samples of RM734 confined between glass plates with polystyrene coatings confirm unequivocally that the observed conic sections do not require any thickness gradients, Fig. 3a,b,c.”

Addressing the wall or line question, we explained in detail the shape of the DW in the newly added paragraph on p.12:

The observed P-, H-, and B-DWs are 3D objects of a discernable length and width (on the order of 10 μm) in the plane (x, y) of the sample; T- and C-walls are narrower, being a few micrometers wide (Supplementary Figure 5). In other words, the DWs are three-dimensional objects as their width is either larger or comparable to the thickness h . An interesting question is whether the DW structure changes substantially along the normal z to the sample. We explore the issue by imposing shear onto the textures, namely, shifting the glass plates of the RM734 N_F cell with polystyrene coatings (Supplementary Movie 2). The shear causes the DWs to shift and extend along the shear direction but they do not split into distinct disclination “lines”, which means that the structures preserve the wall character along the z direction. Similarly, the +1 disclination cores of vortices do not split into two surface point defects. Some variation of the DW and +1 disclinations along the z direction is expected, as in-plane deformations might trigger out-of-plane distortions; known examples are the so-called splay cancelling³⁰, structural twist in confined achiral nematics^{31,32}, and twist relaxation of bend²⁹ mentioned above. The submicron details of the 3D structure of DWs should be explored by means such as electron microscopy since 3D optical imaging by fluorescence confocal polarizing microscopy (FCPM)³³ and coherent anti-Stokes Raman Scattering (CARS)^{34,35} are not reliable because of the high birefringence of DIO¹⁸ and RM734 (Supplementary Fig.2b), which defocuses the probing beam³³.

R1.3. Here are some more minor comments:

1. Some typos and undefined terms. e.g. “... fluid that imposes a degenerate in-plane alignment of the polarization-director $\mathbf{P}\hat{\mathbf{n}}$ couple.” “The T-wall bends the $\mathbf{P}\hat{\mathbf{n}}$ couple parallel to itself,” What is the physical meaning of $\mathbf{P}\hat{\mathbf{n}}$? First time to see this notation and used frequently. Should read \mathbf{P} (and $\hat{\mathbf{n}}/P$), $\mathbf{P}\cdot\hat{\mathbf{n}}$?

A1.3. We are thankful for the comment. We changed the confusing notation $\mathbf{P}\hat{\mathbf{n}}$ to a more reasonable one $\mathbf{P}, \hat{\mathbf{n}}$ by adding a comma between \mathbf{P} and $\hat{\mathbf{n}}$. For example, the cited sentence now reads “The T-wall bends the $\mathbf{P}_1, \hat{\mathbf{n}}_1$ couple parallel to itself.”

R1.4. 2. Loose in references. Some important references are not cited. See above.

A1.4. We thank the Reviewer for the suggestion to expand the list of references. We followed the advice, and added new references, most notably the seminal paper by Landau and Lifshits (1935) which introduced the important concept of surface-generated domain wall structures.

R1.5. 3. Figures should be reorganized. The authors use atypical axis labels. They should reformat the axis label to right positions.

A1.5. We are thankful for the comment and revised the figures as suggested.

R1.6. 4. Units should be written in nonitalic.

A1.6. We are thankful for the comment and revised the units as suggested.

Reviewer #2 (Remarks to the Author):

R2.1. "Ferronematic liquids with Conics" by Kumari et al is a beautiful piece of work, with a comprehensive coverage of the textures formed by this most exciting and current of novel materials when in contact with degenerate surface conditions. There is no hesitation that this work should be published and that Nature Communications is a good choice of journal.

A2.1. The thank the Reviewer for the evaluation of our work. In the revised manuscript, we added new experiments on the second ferroelectric nematic, RM734, placed in cells formed by rigid (glass) plates with degenerate in-plane anchoring, in order to demonstrate the generality of the original findings.

R2.2. Some minor points the authors should consider are listed below.

"find their material realization.." is a little grand. Conic sections are well known to be important in smectic liquid crystals. Chiral smectic C liquid crystals are fluidic and hence may be thought of as ferrofluids. I would recommend being a little less poetic and using "play a central role as intrinsic..." or something similar.

A2.2. We reduced the poetic connotations by rephrasing the statement: "The remarkable bisecting properties of conic sections, known for millennia, play a central role as intrinsic features of liquid ferroelectrics."

R2.3. It would be instructive to include a red-tint plate (or similar lamda plate) for (at least one) of the photomicrographs. For example, it is stated on page 1 that splay is avoided and the domains circular. This is evinced by the Polscope image in Fig 1d). However, the degeneracy should be immediately evident to the reader if the lamda plate photomicrographs were also included.

A2.3. We are thankful for the suggestion and followed it by inserting the following on p.3-4:

"However, polarizing optical microscopy textures show the prevalence of bend; the N_F samples avoid splay and form domains with nearly uniform and circular director fields, as illustrated in Figs. 1,2, Supplementary Figures 3,4 for DIO films and in Fig.3 for flat cells of RM734. The circular director field, representing a disclination – a vortex of a topological charge +1, is directly mapped by the PolScope Microimager (Hinds Instruments) in Figs.1d and 3b. The circular vortices also manifest themselves under a conventional polarizing microscope as Maltese crosses with four extinction brushes, located in the regions where \hat{n} is either parallel or

perpendicular to the linear polarization of incoming light, Figs. 1a,g-i, 2a,c,e, 3a,c. Observations with a full-waveplate 550 nm optical compensator support the circular character of the director by showing interference colors of added retardance in the North-West and South-East quadrants of the Maltese cross (where \hat{n} is parallel to the slow axis of the compensator) and diminished retardance in the North-East and South-West quadrants (where \hat{n} is perpendicular to the slow axis of the compensator) (Supplementary Figure 4).”

R2.4. The jump to describing the different wall structures is a little severe, and left me searching through the paper when I got to the B wall. Everything had been very clear yet simple to that point. Hence, I think it would be better to soften this with an earlier mention that Fig 1 and 2 (2 in particular) have examples of five different types of wall P, H, T, B and C, which will be described in detail in following sections.

A2.4. We follow this suggestion and extended the text before the paragraph describing the P-walls:

” The DWs satisfy Eq.(1) with either $e \approx 1$ (parabolic, or P-walls) or $e > 1$ (hyperbolic, or H-walls) everywhere, except for the tip regions. Near the tips, the fits yield a much smaller e characteristic of elliptical and circular arcs; these arcs are abbreviated as T-walls. The T-walls are 180° DWs, separating two antiparallel polarizations and bounded by two $-1/2$ disclinations. Besides the P-, H-, and T-walls, we also distinguish rectilinear or slightly curved B-walls (with a weak bend of the \mathbf{P} , \hat{n} couple) that separate two closely oriented polarization fields, and C- walls, enclosing central parts of circular vortices. All these DW structures and the mechanisms of their formation are detailed below.”

R2.5. Page 1:

"..forms nearly uniform and circular domains" is not clear from the figures referenced. Presumably, it refers to the photomicrographs of these figures, such as Fig 1a), g), and h), etc. Presumably, it means that the defect at the foci retain orthogonal brushes to some distance, thereby implying circular structure of the director field. It would be helpful to the reader to clarify what is meant.

A2.5. We follow the suggestion to clarify the description and added the abovementioned text A2.3. on p 3 instead of "..forms nearly uniform and circular domains".

R2.6. Seems odd seeing "focuses" alongside "vortices" in the figure caption for Fig 1. I believe foci is preferable, but this might be a journal preference to use contemporary US English.

A2.6. We replaced “focuses” with “foci” and remain open to any editorial suggestions.

R2.7. The labels for Fig 1 b and c are confused. Both are fits to equation 1, with b) the eccentricities and c) the directrices.

A2.7. We clarified the issue by modifying the caption to “**b** The corresponding eccentricities e . **c** The corresponding eccentricities e and distances d to directrices as fits to Eq.(1).”

R2.8. Figure 2 should mention the B wall in the caption.

A2.8. We followed the suggestion and mentioned the B-walls in Figure 2 caption

R2.9. Page 4: " as discussed later".

A2.9. Incorporated.

R2.10. Page 9: the use of Ref before 17 is inconsistent and un-necessary. Move to before the equation if preferred.

A2.10. We followed the suggestion.

Reviewer #3 (Remarks to the Author):

R3.0. Ferroelectric nematic (NF) liquid crystals is a both a new material and new field of very intense research. It has gained large interest far outside the liquid crystal research community.

The authors of the present manuscript have studied a ferroelectric nematic liquid crystal in a novel sample geometry, using degenerate planar and slippery surfaces - glycerol (bottom) and air (top) - as confinement surfaces. With these azimuthally degenerate boundary conditions twist is not required in the LC. Further, in ferroelectric nematics, as a result of the very large value of spontaneous polarization, splay is avoided in order to minimize space charge from divergence in P . The authors have observed and explained that the resulting NF structure primarily contains pure bend deformations, with domain walls (sheets) therefore, importantly, forming conics. The authors, in great detail, theoretically explain their experimental findings. The methods and conclusions are sound and convincing, and the paper is well written.

The authors also point out the analogy with smectic A materials where twist and bend are suppressed and the resulting structure is primarily splayed in the director, here leading to well-known focal conics in the form of disclination lines.

To summarise, this is a beautiful, original, and important study and I certainly recommend the manuscript for publication in Nature Communications.

A3.0. The thank the Reviewer for the evaluation of our work. In the revised manuscript, we added new experiments on the second ferroelectric nematic, RM734, placed in cells formed by rigid (glass) plates with degenerate in-plane anchoring, in order to demonstrate the generality of the original findings.

R3.1. I have only the following (minor) comments/suggestions:

1) In the abstract the authors suggest that:

“The findings could be helpful in designing patterns of electric polarization and space charge”. However, this statement is as far I can see not discussed or elaborated on in the paper. I think such a statement in the abstract should indeed be discussed further/motivated (seemingly being one conclusion from the study).

A3.1. We thank for the suggestion and add an extended explanation at the end of the Main text:

“The demonstrated interplay of electrostatics and geometry that shapes the DWs in ferroelectric fluids could be potentially explored in the design-on-demand of electric polarization

and space charge. For example, surface patterning could be used to predesign a gradient director field with certain deformation types, including splay, which would generate patterns of space charge with a pre-programmed response to an externally applied electric field.”

R3.2. 2) I think it would even be more interesting if the authors could comment on possible implications of the results for the future potential use of the NF materials in e.g. display or photonics applications.

A3.2. We hope that the text above outlines the potential implications.

R3.3. 3) The estimation of the film thickness h could be clarified in little more detail.

In caption Figure 1:

a) “average film thickness is estimated by the interference colors”.

In the text:

b) “Optical retardance Γ of NF films is close to its maximum possible value $h\Delta n$, where Δn is birefringence measured at different temperatures and wavelengths previously. Thus \mathbf{P} is tangential or only slightly tilted away from the xy film plane”.

In Methods:

c) “an average thickness \bar{h} defined by the deposited mass and the area of film.”

My question is simply: Was h calculated from the weight and density of DIO, and the film area (c), or was h calculated from the retardation and birefringence, assuming tangential orientation of \mathbf{P} (which is what you would expect) (a), or was the tangential orientation of \mathbf{P} determined from the birefringence, using a value of h estimated from (c)?

A3.3. We clarified the issue in greater detail in the revised Methods section. Briefly, we use both approaches for DIO films: (i) the thickness is deduced by measuring the deposited mass, density (added Supplementary Fig.7) and area; (ii) the thickness is also estimated through the measurements of birefringence and optical retardance. We find that the two methods yield similar values, which suggests that the director in DIO films is tangential.

R3.4. 4) The font size in Figures 2 and 3 could possibly be made little larger.

A3.4. Thank you, we followed the suggestion and increased the font size.

=====

We hope that the revised manuscript adequately addresses all the points raised by the Reviewers and could be accepted for publication.

Sincerely yours,

Oleg D. Lavrentovich,

On behalf of the authors

REVIEWERS' COMMENTS

Reviewer #1 (Remarks to the Author):

I have no further comments.

Reviewer #2 (Remarks to the Author):

The authors have dealt rather comprehensively with the suggestions and corrections required by all three reviewers. I believe the paper is clearer and certainly worthy of publication in Nature comm.